# QuS: Towards High-Performance EfficientViT on FPGA by Quantization and Streamline Co-Design

## Abstract

Vision Transformer (ViT) has achieved significant success in computer vision, in which EfficientViT is widely used because of its lightweight characteristics. However, EfficientViT is still difficult to deploy on edge devices like FPGA because of its efficiency and accuracy concerns. First, from software perspective, existing quantization approaches fail to consider the inter-channel distribution relationship, which cause significant performance degradation under lower-bit setting. Second, from hardware perspective, current DSP-packing methods struggle to support the diverse kernel sizes and strides of convolutions used in EfficientViT, resulting in redundant computation cycles or bit-width overflow. Moreover, due to the mismatch in data layouts between convolution and linear attention, existing solutions require substantial memory resources for data reordering, which often results in pipeline stalling. In this paper, we propose a Quantization and Streamline Co-Design (QuS) framework for lower-bit EfficientViT deployment on FPGA. It includes three main components: adaptive distribution-aware quantization strategy to provide effective quantization, multi-computing in once packing strategy to improve the DSP-packing efficiency, and low-buffer streamline for linear attention scheme to eliminate pipeline stalling caused by mismatched layout. Experimental results show that our QuS framework achieves over 2200 FPS on EfficientViT, which represents a $3.6\times$ speedup over Jetson AGX Orin and also up to a $24\%$ accuracy improvement under 4-bit quantization.

## 1 Introduction

Recently, Transformer achieves remarkable success in various scenarios, among which Vision Transformer Dosovitskiy (2020); Liu et al. (2021) is one of the representatives in computer vision tasks. However, the computation and memory-intensive attention mechanism hinder its practical deployment on resource-limited hardware. As a lightweight architecture, EfficientViT Cai et al. (2023) leverages a hybrid architecture including depthwise separable convolutions and linear attention, which has been employed as the backbone of many widely used approaches like Ground_DINO Ren et al. (2024), EfficientViT-SAM Zhang et al. (2024), and diffusion models Xie et al. (2024).

Thanks to its inherent efficiency and hardware-friendly design, the deployment of EfficientViT has been investigated by many works Shi et al. (2024); Shao et al. (2024). However, existing works only focus on 8-bit quantization. Lower-bit quantization, which theoretically offers greater acceleration, has not yet been explored. There are three main bottlenecks when deploying lower-bit EfficientViT.

First, from the software perspective, there is still a lack of effective low-bit quantization approaches specifically tailored for EfficientViT. Unlike conventional architectures, lightweight models such as EfficientViT Shi et al. (2024) are particularly sensitive to outliers due to their extensive use of depthwise separable convolutions. Although prior methods attempt to mitigate this issue by migrating outliers from activations to weights, they typically determine the migration extent based solely on per-channel maximum values and a fixed migration strength Xiao et al. (2023). This uniform strategy overlooks the substantial variation in activation distributions across channels. As a result, the activation distribution of different channels after migration may suffer from either under-balancing or over-balancing, leading to large tensor-wise activation range. This will cause suboptimal quantiza-

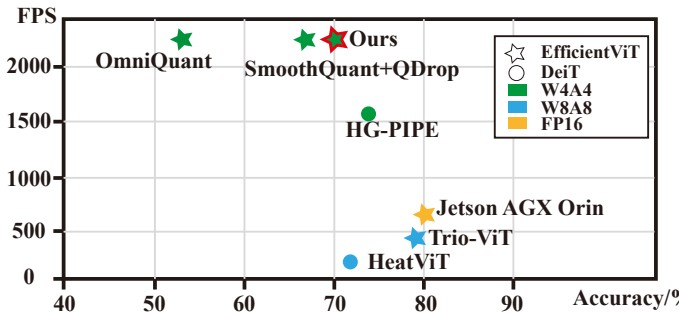

Figure 1: The comparison of accuracy and efficiency of different quantization methods.

tion performance under the hardware efficient tensor-wise quantization scheme, which is commonly used in many approaches Xiao et al. (2023); Shi et al. (2024). This problem is more significant for lower-bit quantization. Therefore, it is desirable to develop a lower-bit quantization approach that explicitly considers the intrinsic data distribution of each channel during outlier migration.

Second, from the hardware perspective, Digital Signal Processor (DSP) packing plays an important role in network acceleration. Existing DSP-packing strategies Liu et al. (2022); Zhang et al. (2023) are designed only for 3×3 convolutions with stride 1, and cannot well support operators in EfficientViT such as 3×3 convolutions with stride 2 (Conv3×3s2) and 5×5 convolutions with stride 1 (Conv5×5s1) under lower-bit settings. Due to the limitation of multiplier bit width (i.e., 27bits×18bits), directly applying existing methods will cause bit width waste for stride > 2 or bit width overflow for kernel size > 3, as illustrated in Fig. 3. Therefore, it is also desirable to design new DSP-packing strategies for lower-bit EfficientViT deployment.

Moreover, the difference in data layouts between convolution and linear attention leads to costly transpose operations, introducing significant memory overhead, stalling the pipeline, and adding extra latency Guo et al. (2024). So, a new linear attention streamline is also required.

To solve the aforementioned problems, in this paper, we propose a systemic software-hardware co-design framework, including an accurate low-bit **Qu**antization algorithm and an efficient hybrid **S**treamline for deploying EfficientViT on FPGA, dubbed **QuS**.

To address the first challenge, we propose **Adaptive Distribution-Aware Quantization (ADAQ)**. Specifically, we introduce Variation Coefficient (VC) as a key statistical metric to quantify the intra-channel activation fluctuation. Instead of relying solely on the maximum value, we leverage the VC to capture the distribution characteristics of each channel, and adaptively determine the outlier migration strength based on the VC information. By considering inter-channel distributional difference, our method provides a more fine-grained control over activation balancing, effectively avoiding the distribution mismatch among different channels for better tensor-wise quantization.

To solve the DSP-packing problem, we propose **Multi-Computing in Once Packing (MuCO)** strategy. Specifically, to support larger strides, we selectively load the required data columns and only perform the computation on selected data for better efficiency. To support larger kernels, we decomposes weight parameters within a kernel into smaller computation groups and perform multiple passes of computation, which effectively prevents multiplication overflow caused by the limited bitwidth of DSP units. Our MuCO DSP-packing strategy is generalizable to arbitrary strides and kernel sizes under 4-bit settings.

In addition, to mitigate the data layout difference between convolution and linear attention, we propose **Low-Buffer Streamline (LBS)** for linear attention. Specifically, we preserve the original data layout and redesign the computation dataflow, to avoid the costly transpose operations.

In summary, our main contributions include:

- We propose a quantization and streamline co-design framework called QuS. To the best of our knowledge, QuS is the first framework to investigate lower-bit quantization and deployment of EfficientViT on FPGA.
- We propose ADAQ strategy, which utilizes both intra-channel statistical information and inter-channel variability for better tensor-wise quantization.
- We design MuCO to support lower-bit convolutions with diverse kernel sizes and strides.

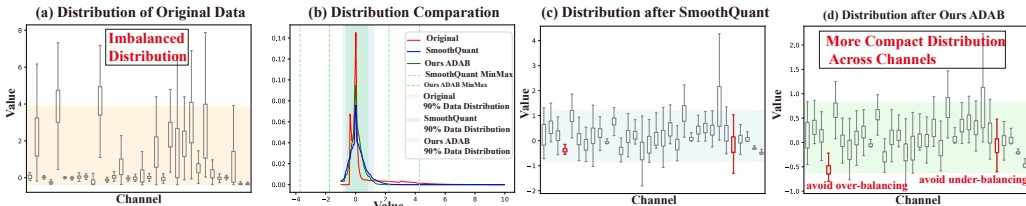

Figure 2: Activation distribution for depthconv in EfficientViT. (a) Original per-channel distribution. (b) Whole tensor distribution of Original, SmoothQuant, and our ADAB. (c) Per-channel distribution after SmoothQuant. (d) Per-channel distribution after our ADAB.

- We propose LBS for linear attention to eliminate memory usage and avoid pipeline stalls.
- Without whistles and bells, our QuS framework achieves over $3.6\times$ speedup compared to Jetson AGX Orin and also up to a $24\%$ accuracy improvement under 4-bit quantization.

## 2 RELATED WORK

**Low-bit Quantization.** Model quantization Nagel et al. (2020; 2019); Wei et al. (2022) is widely applied in the deployment on edge devices, among which DFQ, SmoothQuant, OmniQuant, and Trio-ViT are representative methods. DFQ Nagel et al. (2019) performs equivalent rescaling of the weight ranges across different channels in adjacent layers to achieve a more balanced weight distribution. But it does not support piecewise non-linear activation functions like HardSwish used in EfficientViT. SmoothQuant Xiao et al. (2023) migrates outliers from activations to weights to obtain a quantization-friendly activation distribution. OmniQuant Shao et al. (2023) further introduces learnable equivalent transformations to handle activation outliers. Trio-ViT Shi et al. (2024) adopts channel-wise migration for activations and support depthwise convolution. However, these methods ignore the inter-channel distribution variability, leading to distribution mismatch among different channels and degraded performance. In contrast, our QuS adaptively adjusts the migration strength based on inter-channel distribution variability, to yield more quantization-friendly activations.

**DSP-Packing for Convolution.** Recently, many studies Liu et al. (2022); Sommer et al. (2022); Zhang et al. (2023); Lee et al. (2018); Luo et al. (2023) also focus on the data packing strategy to make full use of Digital Signal Processor (DSP) unit and increase the throughput. For example, Lee et al. (2018) proposed D-MAC for 8-bit quantization and achieve practical speedup. Sommer et al. (2022) designs dsp-packing4 strategy, which packs and computes four 4-bit multiplications on a single DSP in one clock cycle. HiKonv Liu et al. (2022) proposed DSP-packing6, which packs two 4-bit inputs and three 4-bit weights into one DSP. However, all of these approaches are designed for $3\times3$ convolutions with stride 1. The convolution with various kernel size and strides in EfficientViT (e.g., Conv3$\times$3s2) are not supported. So, we design MuCo in our QuS for more efficient inference.

**Accelerator Architecture.** Accelerator architectures are typically classified as fixed or stream-based Chen et al. (2024). Due to the reuse of uniform compute units, fixed architectures Shi et al. (2024); Shao et al. (2024) often rely on frequent off-chip memory accesses for intermediate result storage. In contrast, the stream-based architecture Jiang et al. (2022), customizes computational units for each layer, enabling pipelined dataflow for higher resource utilization and throughput. Therefore, the stream-based architecture is widely studied and adopted. Uint-Packing Zhang et al. (2023) is specifically designed for convolution with stride 1. HG-PIPE Guo et al. (2024), designed for transformers, implements a caching mechanism for attention structure. Since EfficientViT adopts a hybrid architecture of convolution and linear attention, the input layouts for convolution (e.g., NHWC) and attention (e.g., NCHW) differ. The overhead introduced by data layout transformations should be considered. So our QuS introduces LBS strategy to avoid costly transpose operations.

## 3 METHODOLOGY

### 3.1 ADAPTIVE DISTRIBUTION-AWARE QUANTIZATION

**Challenge.** Due to the lightweight depthwise convolution(DW) and pointwise convolution(PW) design, EfficientViT is more sensitive to outliers. Previous methods Xiao et al. (2023); Shi et al.

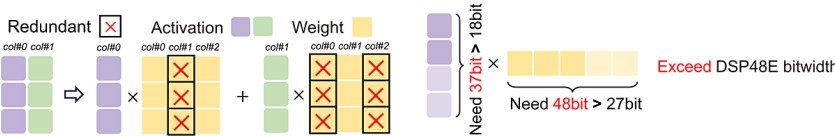

(a) Example of redundant computation for Conv3x3s2      (b) Example of bit width overflow for Conv5x5s1

Figure 3: Motivation of MuCO for convolutions to avoid redundant computation and data overflow.

(2024) reduce activation outliers by migrating them to weights via a balancing factor $\gamma$:

$$Y = \text{fun}\left(W, X\right) = \text{fun}\left(W \cdot \gamma, X/\gamma\right), \tag{1}$$

where, $X$ represents the activation and $W$ represents the weight. $\gamma$ is computed based on the maximum value of activations and weights in each channel, which is written as:

$$\gamma = \frac{\max_{per-channel}(X)^{\alpha}}{\max_{per-channel}(W)^{1-\alpha}}, \tag{2}$$

where $\alpha$ serves as the migration strength, which controls how much of the activation distribution is transferred to the weight. Existing methods use fixed migration strength $\alpha$, but ignore the unique statistical characteristic of each channels as shown in Fig. 2(a). Channels with high variation may be underbalanced, leaving quantization-sensitive outliers; Stable channels may be overbalanced, increasing weight quantization error, as shown in Fig. 2(c).And after applying SmoothQuant, 90% of the data are confined within a narrow range, while the remaining 10% of outliers significantly stretch the overall distribution. This channel-agnostic design leads to suboptimal results, especially in lightweight architectures where per-channel differences are amplified.

To retain DW and PW quantization accuracy at low bitwidth, we propose adaptive distribution-aware quantization to mitigate the unbalanced distribution of activation between channels, and minimize the reconstruction error of weight by optimized approximation.

**Adaptive Distribution-Aware Balancing (ADAB).** Using a fixed $\alpha$ in Eq. 2 fails to account for the distributional differences across channels, potentially resulting in under or overbalanced. To address this, we introduce the Variation Coefficient (VC) as a key metric to measure the fluctuation within each channel. It can be defined as:

$$VC_{per-channel} = \left| \frac{\text{std}\left(X_{per-channel}\right)}{\text{mean}\left(X_{per-channel}\right)} \right|, \tag{3}$$

where $X_{per-channel}$ denotes the set of activation values of each channel, $\text{std}()$ denotes the standard deviation, and $\text{mean}()$ represents the average. A larger VC indicates that the activations in that channel are more dispersed and deviate more significantly from the mean, which implies a higher sensitivity to quantization. Based on this observation, we propose a method to adaptively compute the migration strength $\alpha$ for each channel, enabling fine-grained control over the balancing process:

$$\alpha = \text{clamp}\left(\sigma\left(\kappa \cdot VC_{per-channel}\right), a, b\right). \tag{4}$$

Here, $\sigma(\cdot)$ denotes the sigmoid function, which maps the VC values into the range $(0.5, 1)$. The hyperparameter $\kappa$ controls the sensitivity of the mapping, while the $clamp(\cdot)$ function restricts the resulting $\alpha$ values within the interval $[a, b]$, preventing excessive or insufficient balancing. This formulation allows us to compute the per-channel fluctuation statistics using only a small calibration set, and directly derive an appropriate migration strength $\alpha$ for each channel. As a result, we achieve a more precise per-channel balancing operation, improving quantization accuracy while still maintaining the hardware-friendly per-tensor quantization framework, as shown in Fig. 2(d).

**Multi-level Soft Approximation (MSA).** Since the challenge of activation quantization is migrated to the weights, directly applying AdaRound Nagel et al. (2020), which only allows upward or downward rounding for weight, may limit optimization and lead to suboptimal quantized values. Inspired by DSQ Gong et al. (2019), we utilize the differentiable soft quantization function into the weight calibration:

$$W_q = \frac{\tanh\left(\beta \cdot \left(\frac{W}{s_W} - \left\lfloor \frac{W}{s_W} \right\rfloor - \frac{1}{2}\right)\right)}{2 \cdot \tanh(\frac{\beta}{2})} + \left\lfloor \frac{W}{s_W} \right\rfloor + \frac{1}{2}, \tag{5}$$

where $W = W' + \delta W$, $W'$ and $W_q$ are the full-precision and quantized weights, and $s_W$ is the corresponding scaling factor, $\delta W$ is the parameter to be optimized. $\beta$ is a hyper-parameter that

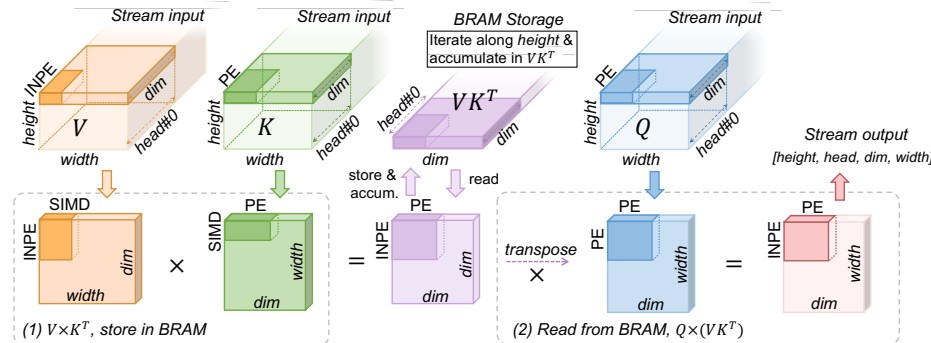

Figure 4: Illustrations of Multi-Computing in Once DSP-packing for different convolutions. (a) Conv3x3, stride 2; (b) Conv5x5, stride 1. Taking one row in feature and kernel as an example.

Figure 5: Low-Buffer Streamline for linear attention, which first calculates $V \times K^\top$ and stores the results in BRAM with a layout of $(head, dim, dim)$, then reads from BRAM and computes the multiplication with $Q$, avoiding huge on-chip memory usage to store the entire $Q, K$ and $V$. Each step multiplies an (INPE × SIMD) matrix with a (SIMD × PE) matrix.

controls the shape and range of the approximation function. It is set to a small value at the beginning of the reconstruction, allowing the weights to be fine-tuned across a wider range rather than being limited to $[0, 1]$. As the reconstruction progresses, we gradually increase $\beta$, making the weights converge to the nearest integers. Thus, weight can be flexibly fine-tuned and gradually converge.

## 3.2 MULTI-COMPUTING IN ONCE PACKING FOR CONVOLUTION

**Challenge.** Existing low-bit DSP-packing methods are mainly tailored for $3 \times 3$ convolutions with stride 1, which limits their applicability to broader architectures. As shown in Fig. 3(a), directly applying these methods to convolutions with stride $S > 1$ leads to inefficiency due to mismatched dataflow. For convolutions with stride $S > 1$, each input column does not need to multiply with all weight elements. However, prior designs assume $S = 1$ and compute all input–weight pairs, resulting in redundant operations when $S > 1$. For kernel sizes that are greater than 3, computing them in a single cycle may exceed the bitwidth of DSP48E. For example, Conv5x5 operator in Fig. 3(b) requires five weights $(w_0, \ldots, w_4)$ for each row of $W$, and four inputs as $(x_0, \ldots, x_3)$ a group Liu et al. (2022) to be loaded. It demands 37bits for activation while 48bits for weights under 4-bit setting(See Appendix for details). While directly separating them into fragments and loading in multiple cycles causes lots of idle bits in DSP and inference latency.

We design a computation paradigm tailored for 4-bit setting to fully utilize DSP resources, named MuCO. It splits the original one-cycle computation process into multi-cycles after once data packing, avoiding redundant computations or overflow. As a general strategy, it is widely applicable to diverse convolutions with various kernel sizes and strides, extending the potential to implement on FPGA.

**For Larger Strides: Selective Fetch.** Assume the input matrix of a convolution is $X = (X_0, \ldots, X_{N-1})$ with $N$ columns, and the convolution kernel is $W = (W_0, \ldots, W_{K-1})$ with kernel size $K$ and stride $S$. We first group the inputs and weights according to their residues modulo $S$: $X_{(s)} = \{X_i \mid i \bmod S = s\}, W_{(s)} = \{W_j \mid j \bmod S = s\}$, where $\forall s \in [0, S-1]$. Therefore,

both the input and the kernel are divided into $S$ groups. Convolution can be performed separately on each of the $S$ groups, resulting in $M$ output groups:

$$Y_m = \sum_{s=0}^{S-1} (X_{(m-s\lceil N/S\rceil)} * W_{(s)}), \forall s \in [0, S-1],\tag{6}$$

where $m \in [0, \lfloor \frac{N-K}{S} \rfloor]$ is the output length. $*$ represents the convolution computation. $\lceil N/S \rceil$ indicates the length of each segment when partitioning a sequence of length $N$ into $S$ subsequences. The final output $Y$ is computed as the shift-and-sum of the $M$ partial results $Y_m$.

Taking Conv3x3 with stride 2 (Conv3x3s2) as an example, the pipelined process is illustrated in Fig. 4(a) by clocks. As the operation of each row is similar, we take one row in convolution operation for illustration. As the kernel slides, we first compute $(x_0, x_2, x_4)$ with $(w_1)$ at Cycle#0, and compute the other three inputs $(x_1, x_3, x_5)$ with $(w_0, w_2)$ at Cycle#1. Then, we accumulate the intermediate results and get 3 outputs. Cycle#2 and Cycle#3 repeat the above computations with the next 6 inputs, and the intermediate result from the last two cycles remains to be added (such as $w_0 x_4$ in Cycle#0 and $w_1 x_5$ in Cycle#1). Therefore, for a convolution with an input size of 256, it only requires $\lceil 256/6 * 2 \rceil$ cycles to get the results in our paradigm, fewer than the original practice ($\lceil 256/2 \rceil$ cycles).

**For Larger Kernel Sizes: Progressive Accumulation.** We further adopt the progressive accumulation strategy. For the general case with $S = 1$, we suppose $K > 3$. Due to the bitwidth limitation of DSP, at most three 4-bit multiplier and two 4-bit multiplicand can be packed. Therefore, for $K > 3$, we split weights into segments following $K = 3A + 2B$, where $A$ and $B$ represent the number of 3-element and 2-element weight segments, respectively. Since convolution can be computed by shifting and accumulating the results of sub-segment convolutions, we divide the input into fragments of length 2. This ensures each 2-element input fragment can be paired with either a 3-element or a 2-element weight segment without exceeding the DSP-packing limit. After computing the dot products between one input fragment and all weight segments, we slide to the next 2-element input fragment and repeat the process. A unified formula for arbitrary strides and kernel size can be written as:

$$Y_m = \sum_{s=0}^{S-1} (\sum_{l=0}^{2-1} (X_{(m-s\lceil N/S\rceil - l\lceil(\lceil N/S\rceil)/2\rceil)} * W_{(s)}))$$

$$= \sum_{s=0}^{S-1} \sum_{l=0}^{2-1} [\sum_{a=0}^{A-1} (X_{(m-s\lceil N/S\rceil - l\lceil(\lceil N/S\rceil)/2\rceil)} * W_{(s-3a)}) \tag{7}$$

$$+ \sum_{b=0}^{B-1} (X_{(m-s\lceil N/S\rceil - l\lceil(\lceil N/S\rceil)/2\rceil)} * W_{(s-2b)})],$$

where $l$ denotes dividing each subsequence of length $\lceil N/S \rceil$ into segments of size 2, and $\lceil \lceil N/S \rceil/2 \rceil$ indicates the total number of resulting segments. The input is processed in groups of two elements. Each 2-element input segment is convolved with $a$ 3-element weight segments and $b$ 2-element weight segments. The results of these partial convolutions are accumulated to produce the final output value.

An illustration of Conv5x5 stride 1 is shown in Fig. 4(b). We divide the kernel size into two groups, and calculate them separately by four cycles. In Cycle#0 and Cycle#1, we first take two input values $(x_0, x_1)$ and multiply them with three weights $(w_0, w_1, w_2)$ and two weights $(w_3, w_4)$, separately. The intermediate results are stored. In Cycle#2 and Cycle#3, we calculate the other two inputs $(x_2, x_3)$ with the five weights, then accumulate and output two results. In this way, it only requires $\lceil 256/2 * 2 \rceil$ cycles with an input size of 256, much fewer than previous practice.

### 3.3 Low-Buffer Streamline for Linear Attention (LBS)

**Challenge.** EfficientViT applies linear attention which eliminates softmax operation:

$$O_i = \frac{\text{ReLU}(Q_i)(\sum_{j=1}^{N} \text{ReLU}(K_j)^\top V_j)}{\text{ReLU}(Q_i) \sum_{j=1}^{N} (\text{ReLU}(K_j)^\top)}. \tag{8}$$

Table 1: Accuracy Comparison of different quantization across various model sizes of EfficientViT.

| Bitwidth | Method | Avg. Bitwidth | b1-r224 | b1-r256 | b1-r288 | b2-r224 | b2-r256 | b2-r288 |
|---|---|---|---|---|---|---|---|---|
| FP32 | - | 32 | 79.39 | 79.92 | 80.41 | 82.10 | 82.70 | 83.09 |
| W8A8QKV8 | SmoothQuant | 8 | 74.91 | 77.80 | 78.07 | 40.00 | 78.01 | 54.75 |
| | SmoothQuant+QDrop | 8 | 78.08 | 79.00 | 79.51 | 81.09 | 80.93 | 80.97 |
| | OmniQuant | 8 | 78.62 | 79.33 | 79.89 | 79.96 | 81.14 | 81.19 |
| | Trio-Vit | 8 | 78.64 | 78.94 | 79.48 | 80.62 | 81.53 | 81.77 |
| | **QuS (Ours)** | 8 | **78.84** | **79.48** | **80.04** | **81.96** | **82.01** | **82.41** |
| W4A4QKV8 | SmoothQuant | 4.7 | nan | nan | nan | nan | nan | nan |
| | SmoothQuant+QDrop | 4.7 | 65.51 | 68.24 | 66.92 | 70.89 | 71.25 | 71.83 |
| | OmniQuant | 4.7 | 43.52 | 53.04 | 44.25 | 65.66 | 69.02 | 69.63 |
| | Trio-ViT | 4.7 | nan | nan | nan | nan | nan | nan |
| | **QuS (Ours)** | 4.7 | **68.22** | **70.07** | **69.99** | **72.50** | **72.92** | **73.46** |

$Q, K, V$ are query, key and value matrices in Linear Attention, $N$ presents the feature dimensions, and $O_i$ denotes the output of the $i$-th row of matrix $O$. Owing to the inherent differences in data layouts between convolution and linear attention computations, the data blocking issue occurs when storing the intermediate results of $V$ and $K$. Following the streamline of Uint-packing Zhang et al. (2023), the data layout of $Q, K$ and $V$ received from the convolution layers follows $(height, channel, width)$, where the number of $channel$ equals $(head \times dim)$. However, the expected data layout in the attention computation is $(head, dim, height \times width)$. Therefore, to read the elements along the $(height \times width)$ dimension, it requires the storage of all $V$ and $K$ matrices, first reshaping them to $(height, head, dim, width)$, and then permuting to $(head, dim, height, width)$, which results in substantial on-chip memory consumption.

We design LBS strategy for linear attention to reorganize data layout during the computation of Attention structure, mitigate the extensive buffer usage of the intermediate outputs. Meanwhile, the computed results can be directly fed into the next layer without transposing, increasing the pipeline throughput by reducing data blocking. The overall process is illustrated in Fig. 5[1]. Our LBS maintains the original data layout of $Q, K$ and $V$ output from convolution layers, and allocates a buffer of $(head, dim, dim)$ in BRAM to store the intermediate results of $VK^\top$ (purple block).

**Step1: $V \times K^\top$**. First, we multiply a $(dim, width)$ slice of $V$ and a $(width, dim)$ slice of $K$ at the head#0. We get $(dim, dim)$ results and store them in the BRAM. According to the data layout of the input stream, we iterate along the $head$ dimension and store the results in BRAM. Then, we iterate along the $height$ and accumulate the output with the previous results in the corresponding index of $head$. In this way, we conduct the multiplication of $VK^\top$ and get a matrix of $(head, dim, dim)$ without storing the entire $V$ and $K$. It only requires a small space for $VK^\top$, which is $(height * width)/dim$ smaller than storing them entirely.

**Step2: $Q \times (VK^\top)$**. Second, we read the $VK^\top$ from BRAM and multiply it with $Q$ matrix. Since the data flow of $V, K$ and $Q$ are input almost simultaneously, $Q$ is also stored in BRAM during the calculation of $VK^\top$. As shown in Fig. 5(2), we read a $(width, dim)$ slice from $Q$ and a $(dim, dim)$ slice from $VK^\top$, resulting in an output of size $(width, dim)$. We iterate the division operation along the $dim$ dimension, and finally iterate along the $head$ and then the $height$ dimensions to collect the output with $(height, head, dim, width)$ layout, which can be input to the next computation unit.

## 4 EXPERIMENT

### 4.1 EXPERIMENT SETUP

**Model and Implementation**. We evaluate the accuracy of the quantized EfficientViT-b series based on the ImageNet dataset Deng et al. (2009) and ADE20K semantic segmentation dataset Zhou et al. (2019). We conducted a comparison of four quantization methods: SmoothQuant Xiao et al. (2023), QDrop Wei et al. (2022) combined with SmoothQuant, OmniQuant Shao et al. (2023) and Trio-ViT Shi et al. (2024). For fair comparison, we compare all quantization methods under both 4-bit and 8-bit settings. For 4-bit setting, we keep Q,K, and V of linear attention in Eq. 8 as 8-bit for better accuracy. The parameter $\kappa$ in Eq. 4 is set to 0.5, so that the $\sigma$ function outputs a value near 0.5

---
[1]For better presentation, we omit the padding to $V$ Cai et al. (2023).

Table 2: The efficiency comparisons of throughput, memory usage, and energy consumptions.

| Metric | GPU Baseline | HeatViT HPCA2023 Dong et al. (2023) | ISCAS 2024 Shao et al. (2024) | Trio-ViT TCAS-I 2024 Shi et al. (2024) | HG-PIPE ICCAD2024 Guo et al. (2024) | QuS (Ours) |
|---|---|---|---|---|---|---|
| Device | Jetson AGX Orin | ZCU102 | ZCU102 | ZCU102 | ZCU102 | ZCU102 |
| Frequency | 930 MHz | 150MHz | 200MHz | 200MHz | 375MHz | 300MHz |
| Architecture | GPU | Fixed-Arch | Fixed-Arch | Fixed-Arch | Stream-based | Stream-based |
| Network | EfficientViT-B1 | Deit-tiny | EfficientViT-B1 | EfficientViT-B1 | Deit-tiny | EfficientViT-B1 |
| Bitwidth | fp16 | W8A8 | W8A8 | W8A8 | W4A4 | W4A4QKV8 |
| Precision | 79.9 | 72.2 | - | 79.3 | 74.4 | 70.1 |
| Method | - | QAT | - | PTQ | QAT | PTQ |
| FPS | 636.9 | 183.4 | - | 447 | 1579 | **2257** |
| GOPs | - | 366.8 | 780.2 | 769 | 3947.5 | 3069.5 |
| Power | - | 9.45W | 7.43W | 7.32W | 21.9W | 20.2W |
| LUTs | - | 137.6k | 104.5k | 130.6k | 212.7k | 198.6K |
| DSPs | - | 1968 | 1024 | 1024 | 78 | 1911 |
| BRAMs | - | 355.5 | 160 | 912 | 324.5 | 470.5 |
| GOPs/kLUT | - | 2.67 | 7.47 | 5.89 | 18.56 | 15.46 |
| GOPs/DSP[1] | - | 0.058 | 0.182 | 0.151 | 0.587 | 0.455 |
| GOPs/W | - | 38.80 | 105.1 | 105 | 180.25 | 151.96 |

[1] Following Shi et al. (2024), we normalize to a DSP-only setting for fair comparison, assuming that one DSP is equivalent to 32 LUTs.

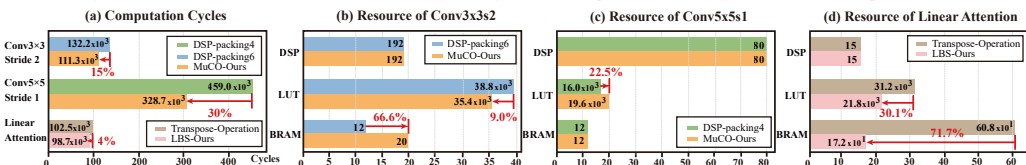

Figure 6: Comparison of (a) computation cycles and (b-d) resource.

when $VC = 1$. The parameters $a$ and $b$ are set to 0.5 and 0.9, respectively. For Eq. 5, the parameter $\beta$ is initialized to 2 and linearly increased to 100 over iterations.

**Architecture and Design**. For the accelerator implementation, we follow the Uint-Packing Zhang et al. (2023); Bao et al. (2021) and utilize Vitis High-Level Synthesis (HLS) for development, deploying on the Xilinx ZCU102 platform at a frequency of 300 MHz. We use our MuCO for convolutions with stride > 1 or kernel size > 3, and use DSP-packing6 for remained conv3×3s1. In addition, Conv1x1 and MatMul are implemented using DSP-packing4 and DSP-packing2, respectively. Linear attention adopts ours LBS design.

## 4.2 END-TO-END EVALUATION

**Accuracy Performance of Low-bit EfficientViT.** The comparison of our method with other methods on the Efficient models is shown in Table.1. Under the W8A8QKV8 setting, Our QuS method reduces the quantization accuracy drop to within $1\%$ across all models, and even below $0.5\%$ in some cases. Under the W4A4QKV8 setting, QuS avoids the severe accuracy degradation observed with SmoothQuant and TrioViT. Compared to OmniQuant, it improves accuracy by nearly $25\%$ on the more compact b1-r224 model. Even against strong baselines combining SmoothQuant and QDrop, QuS achieves an average improvement of $1.86\%$, demonstrating the effectiveness of our method.

**Efficiency Comparison with SOTA Accelerators.** We compared our stream-based FPGA accelerator in QuS with other state-of-the-art implementations on various hardware. Table 2 showcases the settings, capabilities and efficiency performances of different hardware. In terms of throughput, our QuS with the W4A4QKV8 bitwidth setting achieves 2257 FPS, which is $3.5\times$ faster than the GPU baseline of Jetson AGX Orin, and also surpasses the SOTA FPGA implementations by $1.4\times$. The substantial FPS improvement benefits from our optimized MuCO DSP-packing design and the LBS implementation. Additionally, thanks to the optimized DSP packing strategy, our implementation outperforms other works in terms of GOPs/DSP.

## 4.3 ABLATION STUDY

**Effectiveness of ADAQ for Quantization.** We evaluate the individual and combined effects of the two core components in ADAQ through ablation studies (Table. 5). For EfficientViT-b2-224 under the W8A8QKV8 setting, applying ADAB alone to balance activations improves the quantized accuracy by $44.04\%$. Using MSA alone to reconstruct weights yields a $43.01\%$ improvement. Combined, the accuracy increases by $47.99\%$, showing the complementary benefits of both techniques.

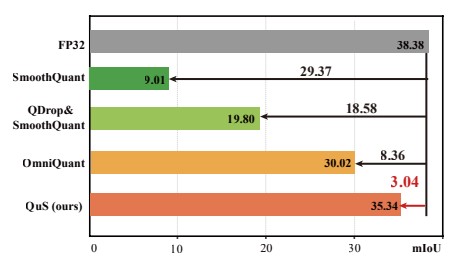

Figure 7: Quantization Accuracy Comparisons of EfficientViT-b1 on ADE20K dataset

Table 3: Quantization and acceleration.

| Quantization on W4A4 setting | | |
|---|---|---|
| model | QDrop | Ours |
| MobileNetV2 | 46.63 | 58.23(**+11.6**) |
| **Acceleration Comparisons** | | |
| model | DSP-packing6 | Ours |
| MobileNetV2 | 151 | 179(**1.19×**) |
| UltraNet | 331 | 520(**1.57×**) |

Table 4: Parameter Analysis of Migration Strength $\alpha$ in the ADAB Strategy of QuS.

| b2-r224 | W8A8QKV8 | W4A4QKV8 |
|---|---|---|
| $\alpha$ | 82.12 | 82.12 |
| 0.25 | 81.20(**-0.92**) | 69.59(**-12.53**) |
| 0.5 | 81.58(**-0.54**) | 71.48(**-10.64**) |
| 0.75 | 81.70(**-0.42**) | 72.11(**-10.01**) |
| ADAB (Ours) | 81.96(**-0.16**) | 72.50(**-9.62**) |

Table 5: Ablation of ADAQ method. ADAB means Adaptive Distribution-Aware Balancing, and MSA means multi-level soft approximation.

| ADAB | MSA | Bitwidth | b2-r224 |
|---|---|---|---|
| - | - | FP32 | 82.12 |
| | | W8A8QKV8 | 33.96 |
| ✓ | | W8A8QKV8 | 78.01 |
| | ✓ | W8A8QKV8 | 76.97 |
| ✓ | ✓ | W8A8QKV8 | 81.96 |

**Generalization Ability of our QuS.** As shown in Fig. 7, we compare QuS approach with other methods under the 8-bit setting on the ADE20K semantic segmentation dataset. Our method outperforms existing techniques on EfficientViT-b1, achieving a 26.33% mIoU improvement over SmoothQuant, and a 5.32% gain over OmniQuant, demonstrating the generality of our approach. As shown in Table. 3, our ADAQ quantization method also benefits MobileNetV2, and MuCo improves deployment FPS by 1.19× on MobileNetV2 and 1.57× on Ultranet Zhang et al. (2023). This further highlights the broad applicability of our QuS framework.

**Efficiency of MuCO for Diverse Convolutions.** In Fig. 6(a)(b)(c), we conduct detailed evaluations of our approach compared with previous DSP-packing methods on a layer with Conv3x3s2 and Conv5x5s1. For Conv3×3s2, our MuCO strategy reduces computation cycles by nearly 15% over DSP-packing6 with similar storage and compute resource usage. The increase in BRAM is negligible compared to the available on-chip resources. For Conv5x5s1, the computation is more compact and efficient compared to DSP-packing4, which reduces about 30% of the computation cycles with a slight increase in LUT resources. This increase is negligible for on-chip resources.

**Efficiency of LBS for Linear Attention.** In Fig. 6(d), taking a single linear attention layer as an example, we compare the resource usage and computation cycles between the attention computation method adopted from HG-PIPE and our LBS design. Our LBS design achieves slightly better computation cycles compared to the Transpose method. In addition, LBS which reduces BRAM usage by 71.7% and LUT consumption by 30.1%. Therefore, LBS significantly alleviates the memory pressure caused by the data transpose.

### 4.4 PARAMETER ANALYSIS

We analyze the migration strength $\alpha$ in Eq. 2, and the results are shown in Table 4. We observe that using ADAB strategy in our QuS framework can outperform the fixed $\alpha$ under different values, which demonstrates the effectiveness of our ADAB strategy.

### 5 CONCLUSION

In this paper, we propose a systemic software-hardware co-design framework by quantization and streamline approaches (**QuS**), achieving accurate and efficient deployment of EfficientViT. It includes an an adaptive distribution-aware quantization method for model quantization and a hybrid structured stream-based FPGA accelerator with Multi-Computing in Once Packing strategy and Low-Buffer Streamline design. Experiments validate the practical performance of deployment for EfficientViT, revealing the potential usage in the real-world edge scenarios.

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

# A APPENDIX

## A.1 QUANTIZATION IMPLEMENTATION DETAILS

All quantization experiments were conducted on a single NVIDIA A800 80G GPU. The initial scale and zero-point for weights were computed using mean squared error (MSE), while those for activations were estimated via an averaged MSE (AvgMSE) over a calibration set of 256 images.

For our ADAQ method, weight reconstruction was performed using a batch size of 32 on a calibration set of 1024 images. The optimization ran for 20000 iterations, with a learning rate of $4.0 \times 10^{-4}$ for weights, and $4.0 \times 10^{-5}$ for both weight and activation scales. The parameter $\kappa$ is set to 0.5, so that the $\sigma$ function outputs a value near 0.5 when $VC = 1$. Because the variation coefficient (VC) of 1 is generally taken to indicate a fairly uniform distribution, we set $\kappa$ to around 0.5 so that the migration strengths of activations and weights remain comparable. The parameters $a$ and $b$ are set to 0.5 and 0.9, respectively. The parameter $\beta$ is initialized to 2 and linearly increased to 100 over iterations. Other hyper-parameters followed the default settings of QDrop.

In our ADAQ method, the balancing process Adoptive Distribution-Aware Balancing(ADAB) can be found in the file $adab.py$, while the Multi-Level Soft Approximation(MSA) is implemented in $msa\_recon.py$.

## A.2 DSP-PACKING METHOD DETAILS

**Block Convolution.** The idea of bloack convolution is to divide a long input sequence into shorter segments whose lengths are comparable to that of the kernel, perform convolution on each segment individually, and then combine the results. Here, we adopt the overlap-add method for aggregation. Taking one-dimensional convolution as an example, given a sequence $h$ of length $N$ and a sequence $g$ of length $K$, their convolution output $y$ can be computed as follows:

$$h[n] = \begin{cases} f[n], & 0 \leq n < N \\ 0, & n < 0 \quad or \quad n \geq N \end{cases}, \tag{9}$$

$$y[m] = (h * g)[m] = \sum_{k=0}^{K-1} h[m-k]g[k], \tag{10}$$

Specifically, assume the input sequence $h$ and kernel $g$ contain $X \times N'$ and $K$ elements, respectively. The input $h$ can be partitioned into $X$ segments, each of length $N'$:

$$h_x = h[xN' : (x+1)N' - 1], \tag{11}$$

The convolution result of $h$ and $g$ can then be computed according to the following equation:

$$y[n] = \sum_{x=0}^{X-1} (y_x[n - xN']), \tag{12}$$

$$y_x[n - xN'] = \sum_{k=0}^{K-1} h_x[n - xN' - k]g[k]. \tag{13}$$

For example, consider two 1D sequences: $h[n] = \{1, 2, 3, 4, 5, 6\}$ and $g[k] = \{1, 2, 3\}$, convolved with a stride of 1. We first divide $h[n]$ into segments of length $N' = 2$: $\{1, 2\}$, $\{3, 4\}$, $\{5, 6\}$. Each segment is then circularly convolved with $g[k]$, result in $\{3, 8, 5, 2\}$, $\{9, 18, 11, 4\}$ and $\{15, 28, 17, 6\}$. Finally, overlapping positions are summed to obtain the final convolution result: $\{14, 20, 27, 32\}$.

**DSP-Packing method.** As shown in Fig. 8(a), the DSP48E2, widely used in FPGAs, supports up to $27 - bit \times 18 - bit$ multiplication. DSP-packing2, as shown in Fig. 8(b), is widely used for 8-bit matrix multiplication. It packs two weights and one input into a DSP to perform two 8-bit multiplications in a single operation. Similarly, as shown in Fig. 8(c), DSP-packing4 is extensively used for 4-bit matrix multiplication, where two weights and two inputs are packed to perform four 4-bit multiplications per DSP operation, significantly improving the multiplication efficiency at 4-bit precision. To fully utilize the bit-width of the DSP under 4-bit convolution, DSP-packing6 packs

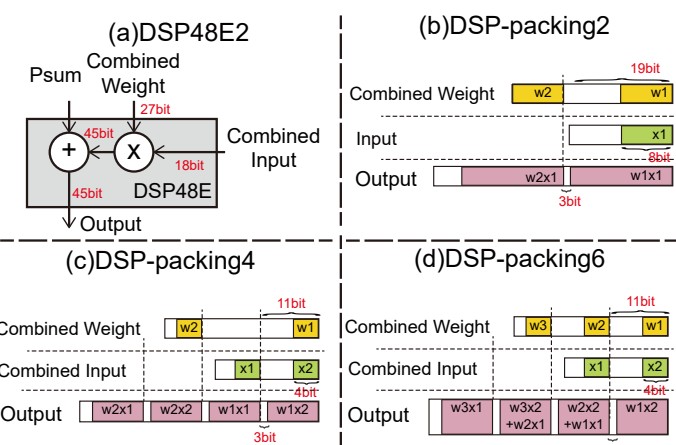

Figure 8: Structure of DSP48E2 and Different DSP-Packing Methods.

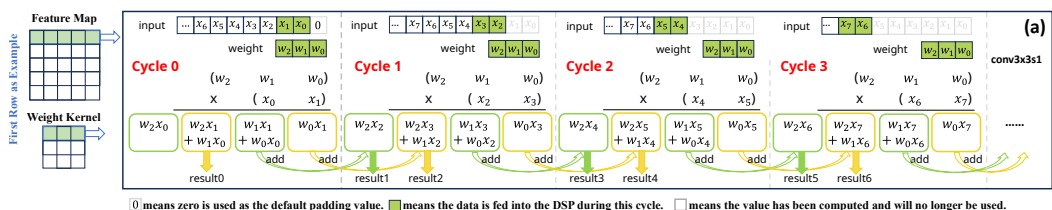

Figure 9: Illustrations of DSP-packing6 for Conv3x3 with stride 1. Taking one row in feature and kernel as an example.

three 4-bit weights into the multiplier input and two 4-bit activations into the multiplicand input. In Fig. 8(d), each segment occupies 11 bits, including 3 guard bits for overflow prevention and 8 bits (from 4-bit weights and 4-bit activations) for the multiplication. Since a 4-bit × 4-bit multiplication produces an 8-bit result, each segment corresponds to a single partial product. As a result, a single DSP multiplication yields a 45-bit output composed of 4 such segments, each representing a circular convolution result of 3 packed weights and 2 packed inputs.

As illustrated in Fig. 9, the computation process of a $3 \times 3$ convolution with stride 1 using DSP-packing6 is detailed for a single row, assuming zero-padding. In the first cycle Cycle #0, the packed inputs $(x_0, x_1)$ and weights $(w_0, w_1, w_2)$ are fed into the DSP for computation. In the next cycle Cycle #1, $(x_2, x_3)$ are packed and processed, and partial results from Cycle #0 are accumulated and output. Subsequent cycles proceed in the same manner, performing convolution by shifting the input window and accumulating intermediate results.

Directly applying the above DSP-packing6 scheme, which is tailored for Conv3×3 with stride 1, to convolutions with stride 2 or larger leads to unnecessary computations and resource waste. For example, if we use the layout in Fig. 9 for a Conv3×3 with stride 2, only $result0, result2, result4, result6, ...$ correspond to valid outputs, while the odd-indexed results are redundant.

For a convolution between a feature map of length 256 and a kernel of size 3, the DSP-packing6 method processes two input fragments per cycle, requiring $\lceil 256/2 \rceil$ cycles in total. In contrast, our proposed MuCO splits the input into even- and odd-indexed sequences, and further divides each into length-3 segments fed into the DSP. As a result, MuCO only requires $\lceil 256/6 * 2 \rceil$ cycles, significantly improving computational efficiency.

For Conv5×5 with stride 1, if five 4-bit weights are to be packed into a single operation, four 4-bit inputs should also be packed to match the typical requirement in block convolution that weight and input fragment lengths be comparable. As a result, the packed weight requires a bit-width of 48 bits: 4 bits for the first (highest) weight, and $4bit + 4 \times (3bit + 4bit + 4bit)$ bits for the remaining four weights, where 3 bits are guard bits and 8 bits ($4bit + 4bit$) are needed to store the

Figure 10: The timing diagram of Linear Attention. Taking $(height, head, dim, width)$=(8,16,16,8) as an example.

Table 6: The efficiency comparisons of throughput, memory usage, and energy consumptions on KV260.

| Metric | SEUer[2] | QuS |
|---|---|---|
| **Device** | KV260 | KV260 |
| **Frequency** | 200MHz | 200MHz |
| **Network** | UltraNet | UltraNet |
| **FPS** | 867 | 1043 |
| **Power** | 5.3W | 5.25W |
| **kLUTs** | 49.3k | 49.2K |
| **DSPs** | 296 | 296 |
| **BRAMs** | 126.5 | 136.5 |

4-bit multiplication result. Similarly, the packed input requires 37 bits: 4 bits for the first input and $4bit + 3 \times (3bit + 4bit + 4bit)$ bits for the remaining inputs. However, the resulting operand size of 48-bit × 37-bit far exceeds the 27-bit × 18-bit multiplication capacity of a single DSP. Therefore, Conv5×5s1 is typically lowered to matrix multiplication and implemented using DSP-packing4, which is more compatible with the DSP's bit-width constraints.

To further improve parallelism, three rows of inputs and three rows of weights can be processed simultaneously in each computation step. Our method, MuCO, similarly supports multi-row parallel computation to enhance throughput.

### A.3 Low-Buffer Streamline Pipeline Analysis

As shown in Fig. 10, we simulated the timing diagram of the linear attention. When the first feature is received, V, K, and Q start processing data simultaneously. Since V and K continuously generate data to be sent to MAT0 for the $VK^T$ multiplication, their processing time is longer than that of Q. Q caches the data and waits for MAT0 to complete before performing the MAT1 operation with MAT0's result. Finally, OUT divides MAT1's result and outputs it. Once the first feature is fully processed, the second follows immediately. The overall process aligns with the design expectations, confirming the effectiveness of our Low-Buffer Streamline design.

### A.4 Performance analysis of Ultranet on kv260 fpga

As shown in Tab. 6, we compare the throughput, memory usage, and energy consumptions of Ultra-Net on KV260 FPGA. Under the same settings, thanks to our MuCO design, our method achieves a $1.2\times$ improvement in inference speed with similar resources.

---

[2]https://github.com/AiArtisan/dac_sdc_2022_champion/tree/master

