# OpenReview forum: "QuS: Towards High-Performance EfficientViT on FPGA by  Quantization and Streamline Co-Design"
_ICLR.cc/2026/Conference — Submitted to ICLR 2026_

### Official Review · Reviewer_iTdT · 2025-10-29

**Soundness:** 2
**Presentation:** 2
**Contribution:** 2
**Rating:** 4
**Confidence:** 4

**Summary:**

This paper proposes the QuS framework for efficiently deploying the EfficientViT model on edge devices such as FPGAs.

To address the accuracy degradation that occurs at low bit depths (e.g., 4-bit), a software-based method called ADAQ is introduced. ADAQ dynamically and effectively migrates activation outliers to weights based on each channel’s variance coefficient (VC) and employs MSA to smoothly optimize weight rounding.

On the hardware side, a novel MuCO DSP-packing strategy is proposed to efficiently support convolutions in EfficientViT with various kernel sizes and strides. This method selectively groups and loads inputs according to the stride, and divides large kernels into segments to prevent DSP overflow while maintaining computational efficiency.

Furthermore, to eliminate the transpose bottleneck caused by mismatches in data layout between convolution and linear attention, an LBS architecture is designed.

Experimental results show that the proposed QuS framework achieves 2257 FPS at 300 MHz on the ZCU102 FPGA board, providing a 3.6× speedup over the Jetson AGX Orin and achieving up to 24% accuracy improvement in 4-bit environments.

These results demonstrate that QuS is an effective software–hardware co-design framework for low-bit quantization and FPGA-accelerated EfficientViT.

**Strengths:**

1. The proposed QuS framework presents a co-design approach that comprehensively optimizes both software and hardware to address the deployment challenges of low-bit EfficientViT models on edge devices. Through this design, QuS minimizes accuracy degradation even under 4-bit post-training quantization (PTQ) settings, while simultaneously achieving a 3.6× speedup compared to Jetson AGX Orin.

2. The proposed ADAQ quantization method introduces a new metric called the Variance Coefficient (VC) for each channel, overcoming the limitations of prior methods that applied fixed-strength migration of activation outliers. By adaptively balancing data distribution, ADAQ effectively mitigates quantization outlier sensitivity in lightweight models using depthwise/pointwise (DW/PW) convolutions, thereby preserving model performance even under low-bit quantization.

3. The proposed MuCO DSP-packing strategy and LBS design provide practical contributions that address major bottlenecks in conventional hardware accelerator architectures. MuCO overcomes the constraints of traditional DSP-packing methods optimized only for conv3×3s1, enabling efficient support for a broader range of kernel sizes and strides. In addition, the LBS design eliminates transpose overhead between convolution and attention modules, resulting in significant on-chip memory savings and improved pipeline efficiency.

**Weaknesses:**

1. In Table 2, the GOPs/DSPs values of HeatViT and HG-PIPE are inconsistent with those calculated from the GOPs and DSPs figures reported in their original papers, making a fair comparison with prior studies impossible. Moreover, although the paper assumes an edge-device deployment scenario, its reported power consumption, LUT utilization, and DSP usage are relatively higher than those of existing works. These issues weaken the authors’ claim that their approach is suitable for edge-device solutions and call for a more detailed analysis.

2. Although the paper mentions that specific critical layers were kept in 8-bit precision under the 4-bit setting, it does not specify which layers were applied or what proportion they represent. Since only the average bit-width is reported, it is unclear whether the observed performance improvement in the 4-bit setting stems from the ADAQ algorithm itself or from retaining 8-bit precision in the critical layers.

3. The authors identify the inter-channel distribution imbalance in EfficientViT as the primary motivation for ADAQ; however, this issue could be more directly addressed through channel-wise quantization. Despite this, the authors rely solely on tensor-wise quantization for hardware efficiency. Yet, they provide no quantitative trade-off analysis showing how much additional hardware overhead channel-wise quantization would incur relative to tensor-wise quantization. The absence of such analysis weakens the claim of the ADAQ algorithm's necessity.

4. The methods employed in the proposed FPGA implementation have already been utilized or discussed in prior accelerator designs, which makes their novelty somewhat limited. Moreover, since these techniques are not specific to EfficientViT and could be applied to other architectures as well, it is somewhat disappointing that the results are presented only for EfficientViT.

5. The paper contains several typos and incorrect figures. In Fig. 4(a), the computation of w0 * x2 is incorrectly shown at Cycle #0, and the corresponding description also presents the wrong operation order. For example, in Appendix A.2, the term “Block Convolution” is misspelled as “bloack convolution.”

**Questions:**

1. Are the GOPs/W and GOPs/DSP values in Table 2 correct? They differ from the values obtained by dividing the reported GOPs by Power and DSP, respectively, so a clarification or verification is needed.

2. Regarding the ADAQ's MSA weight reconstruction step (20,000 iterations), could the authors provide: (1) The accuracy convergence curve plotted against the number of iterations? and (2) The total time required to complete this entire calibration step?

3. Although the paper mentions that certain critical layers were kept in 8-bit precision under the 4-bit setting, the explanation about which specific layers were applied and what proportion they represent is missing. Could you please provide a more detailed explanation of this part?

---

> ### Author Response · Authors · 2025-12-03
>
> __Q1__: In Table 2, the GOPs/DSPs values are inconsistent with those calculated from the GOPs and DSPs figures reported in their original papers.
>
> __A1__: The GOPs/DSP numbers in Tab.2 are directly copied from the original papers (i.e., Tab.2 of HG-PIPE paper). Our QuS can perform better than existing methods.
>
> __Q2__: The reported power consumption, LUT utilization, and DSP usage are relatively higher than those of existing works.
>
> __A2__: Since these works deploy different models and architectures, it is unfair to directly compare the absolute DSP/LUT usage. To ensure fairness, we report the normalized GOPs/DSP metric in table below, which is widely adopted in prior accelerator studies [1]. We also include GOPs/kLUT for a more comprehensive comparison. We have also updated Tab.2 in the revised version.
>
> __Metric__ | __GPU Baseline__ | __HeatViT __ | __ISCAS__ | __Trio-ViT__ | __HG-PIPE__ | __QuS__
> -|-|-|-|-|-|-
> __GOPs/kLUT__ | - | 2.67 | 7.47 | 5.89 | 18.56 | 15.46
> __GOPs/DSP__ | - | 0.058 | 0.182 | 0.151 | 0.587 | 0.455
>
>
>
> __Q3__: The specification on which layers are kept in 8-bit precision under the 4-bit setting.
>
> __A3__: In our linear-attention computation, we keep the Q, K, and V in linear attention at 8-bit precision, while all remaining layers in EfficientViT are quantized to 4-bit. We also evaluate all baseline methods under the same setting for fair comparison. We have clarified this in our revised version.
>
> __Q4__: The authors identify the inter-channel distribution imbalance in EfficientViT as the primary motivation for ADAQ; however, this issue could be more directly addressed through channel-wise quantization.
>
> __A4__: While channel-wise activation quantization could theoretically address inter-channel imbalance, it introduces additional hardware overhead on FPGA. Table below shows the efficiency comparsion of one convolution layer in EfficientViT. The results indicate approximately 35% additional LUT consumption and 26% timing degradation for channel-wise quantization compared with tensor-wise quantization.
> Given these costs, we adopt tensor-wise activation quantization, which is also used in many works [2, 3, 4].
>
> __Metric__ | channel-wise | tensor-wise
> -|-|-
> __FPS__  | 712.1 | 898.4
> __kLUTs__  | 47.7 | 35.4
> __DSPs__  | 204 | 192
> __BRAMs__ | 24.5 | 20
>
> [1] Guo Q, Wan J, Xu S, et al. Hg-pipe: Vision transformer acceleration with hybrid-grained pipeline[C]//Proceedings of the 43rd IEEE/ACM International Conference on Computer-Aided Design. 2024: 1-9.
>
> [2] Shi H, Shao H, Mao W, et al. Trio-ViT: Post-training quantization and acceleration for softmax-free efficient vision transformer[J]. IEEE Transactions on Circuits and Systems I: Regular Papers, 2024.
>
> [3] Xiao G, Lin J, Seznec M, et al. Smoothquant: Accurate and efficient post-training quantization for large language models[C]//International conference on machine learning. PMLR, 2023: 38087-38099.
>
> [4] Zhang J, Zhang M, Cao X, et al. Uint-packing: Multiply your dnn accelerator performance via unsigned integer dsp packing[C]//2023 60th ACM/IEEE Design Automation Conference (DAC). IEEE, 2023: 1-6.

---

> ### Author Response · Authors · 2025-12-03
>
> __Q5__: The methods employed in the proposed FPGA implementation have already been utilized or discussed in prior accelerator designs, which makes their novelty somewhat limited.
>
> __A5__: While prior works have discussed DSP-packing techniques, existing 4-bit DSP-packing schemes are limited to conv3×3s1 and cannot be directly applied to convolutions with larger strides or kernel sizes. Directly applying them to such layers leads to significant DSP underutilization and performance loss. Our MuCO module is the first to support 4-bit DSP-packing to arbitrary strides and kernel sizes, enabling efficient computation for a much broader set of convolution operators. Moreover, existing methods fail to consider the resource consumption of the transpose operation in attention.
> Our LBS can effectively reduce this resource consumption through caching the intermediate representation instead of the whole key and values.
>
> Although our QuS is designed for EfficientViT, we can also apply the hardware implementation on other networks. To further demonstrate the generalization ability of our approach, we additionally apply MuCO to UltraNet, and results are provided in the below table. Experiments show that our method consistently improves performance, confirming that MuCO is not limited to EfficientViT and generalizes well across architectures.
>
> __Metric__ | SEUer[5] | QuS
> -|-|-
> __Device__  | KV260 | KV260
> __Frequency__  | 200MHz | 200MHz
> __Network__  | UltraNet | UltraNet
> __FPS__  | 867 | 1043
> __Power__  | 5.3W | 5.25W
> __kLUTs__  | 49.3k | 49.2K
> __DSPs__  | 296 | 296
> __BRAMs__ | 126.5 | 136.5
>
> __Q6__: The paper contains several typos and incorrect figures.
>
> __A6__: Thank you for pointing this out. We have corrected the figure and fixed typos accordingly in our revised version.
>
> __Q7__: Could the authors provide: (1) The accuracy convergence curve plotted against the number of iterations? and (2) The total time required to complete this entire calibration step?
>
> __A7__: We provide tabular data on MSA weight reconstruction with different numbers of iterations for EfficientViT b1-r224 under W4A4QKV8 setting. Our ADAQ can converge fast during the weight reconstruction process. Our entire quantization process takes only about 4 hours.
>
> __Iter__ | 5000 | 10000 | 15000 | 20000
> -|-|-|-|-
> __Acc__  | 32.8 | 46.2 | 54.6 | 68.2

---

### Official Review · Reviewer_pjAq · 2025-10-31

**Soundness:** 3
**Presentation:** 2
**Contribution:** 3
**Rating:** 6
**Confidence:** 3

**Summary:**

This paper presents QuS, a novel framework for deploying the EfficientViT vision transformer on FPGAs using a software-hardware co-design approach. The novel contribution is a tightly integrated system that pairs a new quantization algorithm with a custom hardware accelerator to enable high-performance inference at low bit-widths (4-bit). The core methodology involves three components: a software-based Adaptive Distribution-Aware Quantization (ADAQ) algorithm that uses a channel's statistical properties to preserve accuracy; a hardware-based Multi-Computing in Once Packing (MuCO) strategy to efficiently process diverse convolution types on FPGAs; and a Low-Buffer Streamline (LBS) hardware design to resolve data layout mismatches between model layers, eliminating pipeline stalls.

**Strengths:**

* The paper is a good example of holistic system optimization. Instead of treating the quantization algorithm and the hardware accelerator as separate problems, it presents an integrated solution where the software (ADAQ) is designed to produce a representation that the custom hardware (MuCO and LBS) can execute with maximum efficiency.
* The QuS framework achieves a throughput of over 2200 FPS, which is a 3.6x speedup compared to a high-end edge GPU like the Jetson AGX Orin. It also demonstrates remarkable accuracy recovery at an aggressive 4-bit precision, a setting where baseline methods completely fail.
* The MuCO strategy for handling convolutions with varied kernel sizes and strides is presented as a generalizable template, not just a solution for EfficientViT. This extends the potential impact of the research to other modern neural network architectures that use unconventional operators.

**Weaknesses:**

* The performance comparison is primarily focused on designs on the same ZCU102 FPGA platform. The paper lacks a broader comparison against newer FPGAs, such as the VCK190. This is a notable omission compared to some baseline frameworks that have published results demonstrating much higher performance on the VCK190 platform, which would have provided a more challenging and relevant benchmark.

* The paper uses the GOPs/DSP metric to claim superior hardware efficiency. However, this metric may not provide a complete picture, as some baseline frameworks use a normalized value for this calculation since their framework utilizes much more LUTs. A direct numerical comparison of GOPs/DSP can be misleading because it doesn't account for the computational work performed by non-DSP logic resources, and comparing a directly calculated value against a normalized one is not an apples-to-apples evaluation. Metrics like GOPs/kLUT, which could offer a more balanced assessment, are also not provided.

**Questions:**

Please refer to the weakness part.

---

> ### Author Response · Authors · 2025-12-03
>
> __Q1__: The paper lacks a broader comparison against newer FPGAs.
>
> __A1__: Thank you for the insightful comment. To better demonstrate the effectiveness of our method, we have also evaluated QuS on a newer FPGA platform (KV260) and new network architecture UltraNet. The results are shown in Table below. From Table below, our QuS achieves substantial performance improvements over baseline methods on KV260 as well. This demonstrates that our approach generalizes effectively across different FPGA generations and network structures. We have added the results in Table 6 of our revised version.
>
> __Metric__ | SEUer[1] | QuS
> -|-|-
> __Device__  | KV260 | KV260
> __Frequency__  | 200MHz | 200MHz
> __Network__  | Ultranet | Ultranet
> __FPS__  | 867 | 1043
> __Power__  | 5.3W | 5.25W
> __kLUTs__  | 49.3k | 49.2K
> __DSPs__  | 296 | 296
> __BRAMs__ | 126.5 | 136.5
>
> __Q2__: The GOPs/DSP metric may not provide a complete picture, as some baseline frameworks use a normalized value for this calculation since their framework utilizes much more LUTs.
>
> __A2__: Thank you for the suggestion. We follow your suggestion and report the normalized GOPs/DSP in Table below. Our QuS still achieves higher GOPs/DSP than other baseline methods and remains competitive with HG-PIPE. The slightly lower GOPs/DSP of QuS compared with HG-Pipe is mainly because HG-Pipe extensively relies on LUT-based approximate computations. While this allows HG-Pipe to achieve slightly higher GOPs/DSP, it requires QAT to recover accuracy, which involves significantly longer fine-tuning. In contrast, QuS uses precise numerical computation and only requires PTQ, enabling a much faster quantization process. We have revised the normalized GOPs/DSP in Table 2 of the revised version.
>
> __Metric__ | __GPU Baseline__ | __HeatViT __ | __ISCAS__ | __Trio-ViT__ | __HG-PIPE__ | __QuS__
> -|-|-|-|-|-|-
> __GOPs/kLUT__ | - | 2.67 | 7.47 | 5.89 | 18.56 | 15.46
> __GOPs/DSP__ | - | 0.058 | 0.182 | 0.151 | 0.587 | 0.455
>
> [1] https://github.com/AiArtisan/dac_sdc_2022_champion/tree/master

---

### Official Review · Reviewer_Lw3w · 2025-11-01

**Soundness:** 3
**Presentation:** 3
**Contribution:** 2
**Rating:** 4
**Confidence:** 4

**Summary:**

This paper presents QuS, a quantization and streamline co-design framework for deploying EfficientViT on FPGA under low-bit (4-bit) precision.
QuS introduces Adaptive Distribution-Aware Quantization (ADAQ), which uses a Variation Coefficient (VC) to measure channel-wise activation variability and dynamically adjust migration strength, improving low-bit quantization fidelity. The authors further propose Multi-Computing in Once (MuCO), a generalized DSP-packing scheme supporting arbitrary kernel sizes and strides, maximizing 4-bit parallelism without overflow. Low-Buffer Streamline (LBS) is proposed to address the shape flow of linear attention.
The speed gain is verified on Xilinx ZCU102 FPGA, and the authors report image classification and segmentation results.

**Strengths:**

1. The paper tackles a practical and relevant problem, i.e., deploying lightweight ViT architectures on resource-limited FPGAs.

2. Results are clearly measured on real hardware with throughput, resource, and power metrics.

3. The proposed pipeline (ADAQ + MuCO + LBS) is cohesive and shows reasonable engineering competence.

4. The experimental section is fairly complete, including ablations and small generalization tests on MobileNetV2.

**Weaknesses:**

1. My biggest concern is novelty. Overall, this work is a bit engineering, and some proposed methods are a bit trivial. ADAQ is essentially SmoothQuant with per-channel adaptation, and the VC-based scaling factor is a simple heuristic. MuCO extends existing DSP-packing methods (DSP-packing4/6, HiKonv) to a few new stride/kernel cases, which is an incremental engineering improvement. LBS removes explicit transposes by keeping the same data layout, but is a routine optimization.

2. This work focuses on EfficientViT, but lacks a good justification for that. The local part of EfficientDiT, i.e., point-wise and depth-wise convolutions, is well established and highly optimized in the literature. On the other hand, for the linear attention part, though it demonstrates appealing results in some recent work, such as SANA, it is still believed to be inferior to softmax attention. In this work, linear attention is the optimization target, but why is softmax attention not good for FPGA? Is higher resolution inference achieved through using linear attention? It is better to show some comparisons and justifications.

**Questions:**

1. The frequency is set up differently in Table 2. How do the claimed gains hold under equal clock frequency and identical resource budgets?

---

> ### Author Response · Authors · 2025-12-03
>
> __Q1__: Concerns about novelty.
>
> __A1__: The primary contribution of QuS is a software–hardware co-design framework. To the best of our knowledge, this is the first framework that enables low-bit (4-bit) PTQ and FPGA deployment of EfficientViT. This has not been achieved in prior literature due to the sensitivity of the quantization on MBConv blocks and inefficiency of linear attention structures.
>
> In terms of technical contribution, we identify the distribution imbalance across different channels severely harms 4-bit PTQ accuracy of EfficientViT, which is not considered by SmoothQuant. ADAQ introduces per-channel adaptive smoothing guided by Variation Coefficient (VC), which we show is crucial for stabilizing MBConv quantization (see Table 1) and achieves usable 4-bit PTQ accuracy of EfficientViT for the first time.
>
> For MuCO, existing DSP-packing methods (DSP-packing4/6, HiKonv) only support stride-1 convolutions. Therefore, prior designs need to utilize conv3×3s1 + maxpool for the downsampling operation [1, 2]. In contrast, our MuCO can generalizes DSP-packing to arbitrary stride/kernel settings, enabling more efficient inference on a wide range of network structures.
>
> Our LBS eliminates large transpose buffers in streaming accelerators, significantly reducing memory overhead for attention. This is a key practicality improvement for deploying EfficientViT on FPGA.
>
> Overall, we design a software-hardware co-design framework. Each module aims to solve the bottleneck of different parts. By integrating all the modules, we are the first to make 4-bit EfficientViT usable and efficiently deploy on FPGA.
>
> __Q2__: This work focuses on EfficientViT, but lacks a good justification for that.
>
> __A2__: Thanks for the suggestion. EfficientViT is a widely used lightweight backbone in many areas (see L38-41 in Introduction). However, there is no existing low-bit quantization method specifically designed for EfficientViT. Although existing works try to optimize the point-wise and depth-wise convolutions in MBConv, they still fail to achieve usable performance under 4-bit setting due to the unbalanced activation distributions across channels. Our work observes this imbalance and designs ADAQ to achieve promising performance under 4-bit setting.
> Moreover, existing 4-bit DSP-packing methods do not optimize for Conv3×3s2 or Conv5×5s1, resulting in underutilized compute resources. Our MuCO design effectively addresses this limitation by improving packing efficiency for these convolution types.
>
> In terms of linear attention, Softmax contains nonlinear operators that are difficult to implement efficiently on FPGA when compared with linear attention. The Softmax operation requires the calculation of exponentiation and normalization (division). And implementing high-precision softmax operators on FPGA requires a large number of complex circuits to be implemented. Therefore, existing methods [3, 4, 5] use lookup table to approximate calculations or iterative approximation circuits to alleviate resource pressure. On the contrary, linear attention does not have this pressure because it is only composed of multiplication and division operators.
>
> [1] Zhang J, Zhang M, Cao X, et al. Uint-packing: Multiply your dnn accelerator performance via unsigned integer dsp packing[C]//2023 60th ACM/IEEE Design Automation Conference (DAC). IEEE, 2023: 1-6.
>
> [2] Luo E, Huang H, Liu C, et al. DeepBurning-MixQ: An open source mixed-precision neural network accelerator design framework for FPGAs[C]//2023 IEEE/ACM International Conference on Computer Aided Design (ICCAD). IEEE, 2023: 1-9.
>
> [3] Guo Q, Wan J, Xu S, et al. Hg-pipe: Vision transformer acceleration with hybrid-grained pipeline[C]//Proceedings of the 43rd IEEE/ACM International Conference on Computer-Aided Design. 2024: 1-9.
>
> [4] Sadeghi M E, Fayyazi A, Azizi S, et al. Peano-vit: Power-efficient approximations of non-linearities in vision transformers[C]//Proceedings of the 29th ACM/IEEE International Symposium on Low Power Electronics and Design. 2024: 1-6.
>
> [5] Wang W, Zhou S, Sun W, et al. Sole: Hardware-software co-design of softmax and layernorm for efficient transformer inference[C]//2023 IEEE/ACM International Conference on Computer Aided Design (ICCAD). IEEE, 2023: 1-9.

---

> ### Author Response · Authors · 2025-12-03
>
> __Q3__: The frequency is set up differently in Table 2.
>
> __A3__: The operating frequencies in Table 2 differ from baseline methods because of the distinct hardware design targets and implementation strategies (e.g., timing closure constraints, target FPGA etc.). We emphasize that even if the clock frequency of baseline method HeatViT and TrioViT are increased to 300MHz, their inference speed is still expected to be significantly lower than our QuS. For example, the FPS will become 670 if we use 300MHz for Trio-ViT method (i.e., 447 for 200MHz will theoretically become 670 for 300MHz).
>
> Moreover, to eliminate the difference under different frequencies, we also report GOPs/DSP and GOPs/kLUT in Table 2 in the revised version, which is widely used metric to measure hardware resource efficiency [1]. Our results still outperform the baseline methods. Note that our approach uses slightly more DSP resources than HG-Pipe. However, HG-Pipe relies heavily on approximate computations to save resource usage and use QAT to recover the accuracy drop, which typically requires several days of fine-tuning. In contrast, our method does not use approximation and only requires 3–4 hours to complete PTQ-based quantization.
>
> __Metric__ | __GPU Baseline__ | __HeatViT __ | __ISCAS__ | __Trio-ViT__ | __HG-PIPE__ | __QuS__
> -|-|-|-|-|-|-
> __GOPs/kLUT__ | - | 2.67 | 7.47 | 5.89 | 18.56 | 15.46
> __GOPs/DSP__ | - | 0.058 | 0.182 | 0.151 | 0.587 | 0.455
>
> [1] Guo Q, Wan J, Xu S, et al. Hg-pipe: Vision transformer acceleration with hybrid-grained pipeline[C]//Proceedings of the 43rd IEEE/ACM International Conference on Computer-Aided Design. 2024: 1-9.

---

### Official Review · Reviewer_K7fS · 2025-11-03

**Soundness:** 2
**Presentation:** 3
**Contribution:** 2
**Rating:** 4
**Confidence:** 4

**Summary:**

This article proposes a quantization and an accelerator co-design for low-precision EfficientViT on FPGAs. Based on channel distribution variation, the authors propose the Adaptive Distribution-Aware Quantization (ADAQ) to improve the existing SmoothQuant approach, which adjusts hyper-parameters based on a metric called Variation Coefficient. In addition to that, they propose a fine-tune method, Multi-level Soft Approximation (MSA), to facilitate weight quantization. In hardware design, to satisfy the architecture requirement of EfficientViT, they generalize existing DSP-packing methods for larger strides and kernel sizes, and propose a method called Low-Buffer Sreamline for Linear Attention (LBS) to re-organize tensor shape and avoid pipeline stalls. In experiments, the proposed method achieves better accuracy and hardware performance compared with existing quantization and FPGA accelerator counterparts.

**Strengths:**

a) ADAQ improves existing SmoothQuant method by adapting the hyper-parameter according to channel distributions. This insight sounds reasonable and novel. Their profiling results in Fig. 2 can demonstrate this insight. It might be better if the author could provide more experiments to showcase how much ADAQ can improve SmoothQuant in terms of data distribution.

b) Multi-Computing in Once Packing (MuCO) generalizes the existing methods for larger strides and kernels.

c) LBS significantly reduces the BRAM cost for implementing the transpose operation in linear attention. The experiment in Fig. 6 (d) indicates 71.7% BRAMs are saved.

**Weaknesses:**

a) The Multi-level Soft Approximation (MSA) method appears abruptly in this manuscript. The authors failed to sufficiently explain its role in the entire quantization method, nor the method itself. They could consider clarifying its connection with ADAQ, and also explain the equation with more details.

b) Fairness of accuracy comparison. The authors compare QUS with SmoothQuant, QDrop, OmniQuant, and Trio-Vit. All of them are post-training quantization methods. However, QUS introduces fine-tuning. This may not be fair for other quantization methods. The authors never mention post-training quantization in manuscript nor clarify the fairness of their accuracy experiments.

c) Potential mistake in Eq. 4. VCs are positive values. In this case, the output of the sigmoid operator will be (0.5, 1). According to the context, the output range is supposed to be in (0, 1).

d) Inaccurate title. The authors call QUS a quantization and streamline co-design. However, it seems like the quantization and hardware design are completely independent. This name might lead to confusion.

**Questions:**

a) QUS seems to introduce a fine-tuning process. However, it is compared with post-training quantization methods in terms of accuracy. The authors could clarify the connection between their methods and PTQ, and also explain why it is fair to compare it with other PTQ methods instead of those with fine-tuning or training.

b) Why is MSA introduced? And how does MSA address the problem?

c) The VCs are positive values. This will force the output of the sigmoid operator in Eq. 4 to be in (0.5, 1). Is this the expected output or an error?

d) Is there any dependence or connection between the proposed quantization algorithm and hardware design? Why is it called a co-design?

e) In experimental setup, some of the critical layers are kept in 8-bits. It would be better to indicate what they are explicitly.

f) In line 291, the ‘k’ in ‘k=3A+2B’ should be uppercase. Please double check similar errors.

g) For references that have been published, it would be more appropriate to cite them directly from their official publishers instead of arXiv. For example, ‘Hg-pipe: Vision transformer acceleration with hybrid-grained pipeline’.

---

> ### Author Response · Authors · 2025-12-03
>
> __Q1__: The Multi-level Soft Approximation (MSA) method appears abruptly in this manuscript \& Why is MSA introduced?
>
> __A1__: Following the standard PTQ pipeline, our ADAQ consists of two stages. The first stage is parameter smoothing, where we adopt ADAB to balance the distribution across channels. The second stage is weight reconstruction, where we use MSA to rebuild the parameters. Therefore, both ADAB and MSA are integral components of our overall ADAQ quantization method.
> Existing methods such as AdaRound only optimize $S_w$ and $\delta W_q$ during the reconstruction stage, following:
>
> $ W_q = \left\lfloor \frac{W}{s_W} \right\rfloor + \delta W_q, $
>
> where $\delta W_q$ is constrained to 0 or 1. This forces each weight to be either rounded up or down, resulting in a limited solution space.
> In contrast, our MSA introduces a smooth reconstruction function. Let $Z=W/S_w$, then, our MSA can be written as shown:
>
> $ W_q = \frac{\mathrm{tanh}\left(\beta \cdot \left(\frac{W}{s_W} - \left\lfloor \frac{W}{s_W} \right\rfloor - \frac{1}{2}\right)\right)}{2\cdot\mathrm{tanh}(\frac{\beta}{2})} + \left\lfloor \frac{W}{s_W} \right\rfloor + \frac{1}{2} \\
>         = \left\lfloor \frac{W}{s_W} \right\rfloor + \frac{\mathrm{tanh}\left(\beta \cdot \left(Z - \left\lfloor Z \right\rfloor - \frac{1}{2}\right)\right)}{2\cdot\mathrm{tanh}(\frac{\beta}{2})} + \frac{1}{2}. $
>
> In this equation, $(Z-\left\lfloor Z \right\rfloor-1/2) \in (-1/2,1/2)$. By applying a tanh⁡-based approximation with parameter $\beta$ on the term $(Z-\left\lfloor Z \right \rfloor - 1/2)$, we can gradually approximate the hard rounding function in the reconstruction process for better performance.
>
>
> __Q2__: Fairness of accuracy comparison \& QuS seems to introduce a fine-tuning process.
>
> __A2__: We would like to clarify that our method strictly follows the standard post-training quantization (PTQ) pipeline and does not introduce additional fine-tuning. We believe the ``fine-tuning process'' in our QuS mentioned by the reviewer refers to the weight-reconstruction stage using the calibration dataset. This process is a standard operation in PTQ and also present in our baseline methods in Table 1 including SmoothQuant+QDrop, OmniQuant, and Trio-ViT.
>
> __Q3__: Potential mistake in Eq. 4 \& VCs are positive values.
>
> __A3__: We thank the reviewer for pointing out this issue. We have revised the manuscript to correct this typo and clarify the intended behavior of Eq.4.
>
> __Q4__: Dependence between the quantization algorithm and hardware design? \& Why is it called a co-design?
>
> __A4__: Thanks for you question. Quantization and hardware are not independent components in our framework. From the software perspective, our ADAQ method improves the 4-bit EfficientViT PTQ accuracy to a usable level, which provides the essential accuracy foundation for the entire pipeline. From the hardware perspective, we propose the MuCO and LSB specifically designed for 4-bit EfficientViT, which can fully leverage the potential speedup of 4-bit compact model and this is not achieved by any existing methods.
>
> __Q5__: In the experimental setup, some of the critical layers are kept in 8-bits. It would be better to indicate what they are explicitly.
>
> __A5__: In our implementation, we keep the Q, K, and V of linear-attention in Eq.8 at 8 bits, while all remaining layers are quantized to 4 bits. We have explicitly clarified this detail in the revised manuscript (See L375-376 in Experiment Setup of revised version).
>
> __Q6__: In line 291, the ‘k’ in ‘k=3A+2B’ should be uppercase. Please double check similar errors.
>
> __A6__: Thanks for your comment. We have corrected this typo in our revised version and proofreading the whole manuscript.
>
>
> __Q7__: For references that have been published, it would be more appropriate to cite them directly from their official publishers instead of arXiv.
>
> __A7__: Thank you for the suggestion. We have updated the reference in our revised version.

---

### Author Response · Authors · 2025-12-03

Dear PCs, SACs, ACs,

We sincerely appreciate the reviewers’ time and constructive feedback. Below we provide a summary of our contributions and highlight the clarifications and improvements made during the rebuttal period.

__Claim of Contribution__

This paper introduces QuS, the first software–hardware co-design framework enabling 4-bit post-training quantization (PTQ) and efficient FPGA deployment of EfficientViT. Our technical contributions include:

- ADAQ, an activation–distribution–aware quantization method. Our ADAQ includes two parts: (1) The ADAB method, which addresses the severe inter-channel imbalance in MBConv and (2) The MSA method, which is a differentiable and annealed soft-rounding module to improve the weight reconstruction performance. For the first time, our ADAQ enable usable 4-bit PTQ accuracy on EfficientViT without fine-tuning.
- MuCO, the first 4-bit DSP-packing method to support arbitrary kernel sizes and strides. This design substantially improves hardware utilization and throughput of EfficientViT and also wider range of network architectures.
- LBS, a lightweight dataflow design that eliminates expensive transpositions and reduces memory overhead for attention in streaming accelerators.

Together, these components form the first solution for accurate 4-bit quantization and efficient FPGA deployment of EfficientViT.

__Summary of Reviews and Responses__

1. Novelty concerns (Reviewer Lw3w, Reviewer iTdT).

    To the best of our knowledge, this is the first framework that enables low-bit (4-bit) PTQ and FPGA deployment of EfficientViT. We would like to clarify the points below:

    - 4-bit EfficientViT PTQ was previously unsuccessful due to MBConv activation imbalance, which is not considered by existing methods. To this end, we design ADAQ to specifically resolves this accuracy bottleneck.
    - Existing DSP-packing methods (DSP-packing4/6, HiKonv) only support stride-1 convolutions. MuCO is the first DSP-packing solution supporting arbitrary stride/kernel settings, which enables more efficient inference on a wide range of network structures.
    - Existing methods need to cache key and value in BRAM, which requires large memory resource. LBS eliminates large transpose buffers in streaming accelerators, significantly reducing memory overhead for attention. This is a key practicality improvement for deploying EfficientViT on FPGA.

2. Fairness of quantization comparisons (Reviewer K7fS).

    We emphasized that QuS is pure PTQ method without involving any fine-tuning process. This pipeline is exactly the same as the strong baseline SmoothQuant + QDrop, OmniQuant and Trio-ViT. All baselines (except SmoothQuant) only use weight reconstruction. So the comparisons remain fair. Detailed responses are shown in A2 for Reviewer K7fS.

3. Edge-device suitability and resource comparison (Reviewer Lw3w, Reviewer pjAq, Reviewer iTdT).

    We have revised our GOPs/DSP value to the normalized one and also additional report GOPs/kLUT for better comparison. Detailed responses are shown in A3 of Reviewer Lw3w. We also add these results in the revised version (Table 2).

4. Clarification of MSA and Eq.5 (Reviewer K7fS).

    We have provided the detailed explanation of MSA, including its motivation, mathematical behavior, and role in our QuS. Detailed responses are shown in A1 of Reviewer K7fS.

5. Clarification of 8-bit layers (Reviewer K7fS, Reviewer iTdT).

    Only Q, K, V in linear attention remain at 8-bit and all other layers are 4-bit. All baseline methods are evaluated under the same setting. Detailed responses are shown in A5 of Reviewer K7fS. We also clarify this in our revised version (L375-376).

We thank the ACs and reviewers again for their valuable feedback. We believe that our clarified analysis, additional experiments, and updates address the raised concerns and further strengthen the contributions and validity of our work.

Best regards,

Authors of submission 3865

---

### Meta-Review · Area_Chair_roti · 2026-01-03

**Summary:**

This paper proposes QuS, which co-designs 4-bit PTQ plus FPGA streamlining for EfficientViT using ADAQ, MuCO DSP-packing, and LBS.

**Reviewer Concerns:**

Main concerns from reviewers include:

1. Limited novelty/incremental engineering, unclear co-design coupling, and questionable/unclear comparison fairness (PTQ vs “fine-tuning” weight reconstruction, mixed precision with Q/K/V at 8-bit).

2. Hardware comparisons raise metric inconsistencies (GOPs/DSP), frequency/resource-budget mismatch, plus missing justification and presentation/typo issues.

**Reviewer Scores:**

Reviewer / Score

K7fS	4

Lw3w	4

pjAq	6

iTdT	4

Average	4.5

No reviewers indicated to increase or decrease their scores.

---

### Decision · Program_Chairs · 2026-01-26

Reject